# Cognitive-behavioral correlates of pupil control ideology

**Elena Mirela Samfira[1,2], Florin Alin Sava[2]***

**1** Teacher Training Department, Banat University of Agricultural Sciences and Veterinary Medicine, Timisoara, Romania, **2** Department of Psychology, West University of Timisoara, Timisoara, Romania

* florin.sava@e-uvt.ro

**Data Availability Statement:** Data is held in the Open Science Framework public repository with a permanent doi: 10.17605/OSF.IO/A8HGJ.

**Funding:** The author(s) received no specific funding for this work.

## Abstract

Teacher's pupil control ideology is a central feature for the quality of the teacher-student relationship, which, in turn, impacts the teacher's level of well-being. The pupil control ideology refers to a teacher's belief system along a continuum from humanistic to custodial views. Teachers with humanistic orientation view students as responsible and, therefore, they exert a lower degree of control to manage students' classroom behaviors. Teachers with a custodial orientation view students as untrustworthy and, therefore, they exert a higher degree of control to manage students' classroom behaviors. The relationship between pupil control ideology and dysfunctional beliefs originated from the cognitive-behavioral therapy framework has not been investigated, despite existing evidence suggesting that the pupil control ideology is linked to stress and burnout. One hundred fifty-five teachers completed a set of self-report questionnaires measuring: (i) teacher's pupil-control ideology; (ii) perfectionistic and hostile automatic thoughts; (iii) irrational beliefs; (iv) unconditional self-acceptance; (v) early maladaptive schemas; and (vi) dimensions of perfectionism. The result suggests that teachers who adopt a custodial view on pupil control ideology endorse more dysfunctional beliefs than teachers who adopt a humanistic view. They tend to present a higher level of perfectionism, unrelenting standards, and problematic relational beliefs, including schemas of mistrust and entitlement. They also present more often other-directed demands and derogation of other thoughts. Such results picture a dysfunctional view on pupils who misbehave, as adversaries who threaten their rigid and/or perfectionistic expectations.

## Introduction

The literature on the teacher-student relationship has mostly focused on the impact of this relationship on children. Much less is known about how this relationship affects teachers and their lives. Classroom disturbance, disciplinary problems, and the emotional involvement of teachers with their pupils [1] are primary reasons for including teaching among the most stressful occupations [2] and among the high-risk occupations for mental disorders [3, 4]. The quality of teacher-student interaction fully mediates the relationship between teachers' perception of pupil misbehaviors and teachers' well-being [5]. The result was replicated [6],

**Competing interests:** The authors have declared that no competing interests exist.

emphasizing that the quality of teacher-student interaction could be a key factor for the well-being and the mental health of teachers [7].

The present research aims at examining how pupil control ideology of teachers [8, 9], a core construct for the quality of the teacher-pupil relationship, is correlated with various cognitive variables from cognitive-behavioral psychotherapies. In the introductory part, we will first provide theoretical and empirical rationales for our focus on the pupil control ideology. Next, we will briefly introduce various dysfunctional beliefs for the readers who are less familiar with cognitive-behavioral interventions. We will provide the rationale for linking the pupil control ideology to these dysfunctional beliefs.

## The pupil control ideology

The teacher-student relationship is conceptualized "as the generalized interpersonal meaning students and teachers attach to their interactions with each other" [10, p.364]. In the students' view, an effective, and supportive teacher-student relationship involves trust, together with warmth, and low negativity apart from the teacher [11].

The pupil control ideology (PCI) [8, 9] is at the core of the teacher-student relationship, determining whether such an interaction is depicted positively or negatively. The pupil control ideology gives meaning to the interaction between teachers and students [9]. It could be defined by "the amount of control teachers believe they should exercise in order to manage students' behavior" [12, p. 3], an aspect that derives from the amount of trust teachers have in their students. PCI quantifies a teacher's belief system along a continuum from humanistic (low scores) to custodial view (high scores). On one end, there is the humanistic view in which teachers perceive pupils as having all kinds of needs that need to be nurtured. Children's misbehaviors are treated as instances of unmet children's needs rather than in moralistic terms. Such behaviors require open communication to solve the unmet needs rather than the use of punitive actions. On the other end, there is a custodial view in which pupils' misbehaviors are seen as signs of disrespect for the teacher, as an affront to the teacher's authority. The preservation of order is the primary concern, often via punitive actions.

Past research showed that the custodial view is the strongest predictor for a teacher conflict-inducing attitude towards pupils [13], and it correlates strongly with a positive attitude towards corporal punishment [14] and with student's perception of classroom environment as boring and dull [15]. These negative outcomes undermine student's trust in teachers and schools [13, 16]. Likewise, a custodial view on pupil control correlates negatively with teaching efficacy [17], with academic optimism [18], and with trust [18, 19], all these constructs having a significant impact on students learning. Overall, "the PCI sets the tone and establishes an atmosphere in the class," being a crucial component for establishing the quality of the teacher-student relationship [18, p.33].

Whereas the over 50-year old educational construct—the pupil control ideology–sufficiently proved its utility in the educational sciences field, there is less information on how it relates with psychological constructs that are relevant from the mental health needs of the teachers. The few existing studies indicate that PCI (custodial view) was positively correlated with the level of occupational stress, e.g. [20], as well as with the burnout level, e.g. [21], and it was negatively related to a mindfulness view, e.g. [22]. Likewise, the custodial view was positively correlated with teachers' authoritarianism [17], and negatively correlated with the Big-Five trait of openness [23]. However, there was no significant association between the PCI and the neuroticism level [23].

The PCI relevance for mental health is doubled since the construct is not only relevant for teachers, but also for the pupils with whom teachers interact. For instance, a custodial view on

pupil control was related to more emotional and somatic complaints apart from the pupils [13], and also led to a lower level of pupil self-actualization and a higher level of student alienation [24].

Teachers with a custodial view on their pupil control ideology are more prone to punish students who misbehave and are more inclined to believe that strict discipline is the key to success. They demand control from their students and blame them for deviating from the strict rules they have imposed. One might expect that such individuals might also endorse more beliefs and attitudes that are detrimental not only for their teacher-student relationship but also for their mental health status and well-being. Such cognitive distortions might positively correlate with a custodial view, as previous studies showed that a custodial view is linked to stress [20] and burnout [21], and inversely linked to mindfulness [22]. Since distinct themes are related to different kind of psychopathology (e.g., hopelessness, loss, and personal failure are often linked to depression), the most interesting for us was to explore the constellation of cognitive distortions that correlates with a custodial view in the pupil control ideology.

## A brief overview of dysfunctional beliefs

Beck [25] introduced his cognitive model for psychopathology in which he states that emotional sufferance is a result of three different layers of cognitive distortions: automatic thoughts (concrete, specific, surface), dysfunctional intermediate beliefs, and dysfunctional core beliefs (abstract, global, deep). Automatic thoughts refer to what goes through one's mind when confronted with a specific situation (e.g., "I cannot handle this student"). Intermediate beliefs refer to rules, attitudes, or assumptions that are cross-situational and influence our interpretation and response to a class of situations (e.g., "If you treat students gently, they will start disobeying you"). Core beliefs refer to the underlying views on the self, others, and the world. They are general, rigid, and more abstract (e.g., "I am weak", "The world is an unsafe place"). The three layers of cognitions interact with each other in a complex way, and cognitive interventions aim at identifying these dysfunctional beliefs and alter them into functional ones [25]. To complicate the things further, in addition to the above distinction (e.g., surface vs. deep cognitions), one could differentiate between cold and hot cognitions [26–28]. Cold cognitions are representations and assumptions of a given fact (e.g., "Nowadays students are less respectful with their teachers"). Hot cognitions refer to the way people further process these cold cognitions, in an evaluative way ("This is terrible"). Irrational beliefs [29] are dysfunctional beliefs, but they mainly involved hot cognitions, referring to the way people appraise the cold cognitions [30]. For instance, a teacher might have a dysfunctional intermediate cold belief that "Students who do not listen are insolent and disrespectful with their teacher". This statement could be functional (true) or dysfunctional (false) from a cold cognition view. However, the same teacher would react differently when endorsing an irrational (hot) cognition such as "This is intolerable" vs. when endorsing a rational (hot) cognition such as "I accept it, as no one is perfect". As expected, such beliefs (e.g., irrational beliefs) are significant predictors of teachers' burnout [31].

## The present study

In the present study, we examined whether teachers' pupil control ideology is linked to specific cognitive distortions. We have no empirical evidence whether the pupil-control ideology (that can be seen as profession-specific, pupil control beliefs) are functional or dysfunctional. However, based on the theoretical analysis presented above and on the empirical evidence linking the pupil control ideology to stress and burnout, it is more likely to find a positive association between the custodial view and specific dysfunctional beliefs.

Addressing this research question is relevant for both theoretical and practical reasons. From a theoretical perspective, the study contributes to enriching the literature on pupil control ideology underpinnings. As already underlined, most previous studies have focused on antecedents and consequences of pupil control ideology from an educational science perspective. There were significantly fewer studies that aimed at addressing this topic from a psychological perspective. Teachers strive to achieve a sense of competence and success. However, their efforts are often undermined by the lack of personal control and unpredictability in a classroom context, leading to stress or burnout. Most psychological studies have focused on depicting custodial teachers' personality profile (e.g., more authoritarian, less open, etc.). Such findings could help determine who the teachers at-risk for manifesting a problematic teacher-student interaction are. However, they are less useful for an intervention purpose because personality traits are stable and difficult to change. Such findings are also less relevant to the aim of decreasing the level of stress among teachers. That is why examining the link between dysfunctional beliefs and the pupil control ideology is of more practical value. The core of any cognitive-behavioral intervention consists of replacing dysfunctional beliefs with more functional ones. Such interventions could provide a new avenue to alter pupil control ideology indirectly by changing the correlated dysfunctional beliefs. Changing teachers' cognitions on aspects such as perfectionistic cognitions or their demandingness beliefs will most likely influence their interaction with students by altering the amount of control they exert in the classroom context. Likewise, addressing teachers' cognitions that could impair interpersonal interaction, such as hostility or entitlement, will change teachers' beliefs about pupils with whom they interact daily, towards perceiving a less threatening classroom environment. Thus, our study opens an avenue towards potential cognitive-behavioral interventions in the school setting by first gathering evidence on the association between the pupil control ideology and specific dysfunctional beliefs. Such interventions have the potential of leading to multiple direct and indirect benefits regarding the classroom climate, classroom management, and teachers and pupils' well-being.

Given the study's potential relevance for cognitive-behavioral interventions, we organized our analysis on three different cognitions levels: automatic thoughts, intermediate beliefs, and schemas as core beliefs.

First, we focus on automatic thoughts. They represent the first thoughts that go to someone's mind in response to a situation. Teachers with a custodial view manifest a high need for control of the classroom environment (e.g. My work should be flawless"). They are also highly reactive to disruptive behaviors that affect their control of the environment (e.g. "What an idiot"). We decided to focus on automatic thoughts about perfectionism [32] and hostility [33]. Both automatic thoughts measures reveal the frequency of thoughts on relevant themes, namely themes of perfection and imperfection for the perfectionism case, and themes of revenge, aggression, or derogation of others for the hostility case. Whereas automatic thoughts could be adapted to consider the context (e.g., in the classroom), we relied on the standard instruction for collecting the self-reported automatic thoughts. The standard instruction asked participants to say how frequently they had a specific thought in the last two weeks, in general, not limited to the classroom context. This way, we ensured a context-free consistency in the measurement of all kinds of dysfunctional cognitions.

H1A. *We expect that teachers with a custodial view on pupil control ideology will experience more often perfectionistic automatic thoughts.*

H1B. *We expect that teachers with a custodial view on pupil control ideology will experience more often hostile automatic thoughts.*

The next level of investigation focuses on the association between the pupil control ideology and irrational beliefs. The latter construct originates from the Rational Emotive Behavior Therapy (REBT) approach [29] and represents intermediate dysfunctional beliefs. Several types (processes) of irrational beliefs have been proposed: *demandingness beliefs*, which refer to inflexible standards and demands about how the world must be / self must behave / the other people must behave; *low frustration tolerance beliefs*, which refer to undesirable results or situations that cannot be tolerated; *self-downing beliefs*, which refer to negative global evaluations of themselves; *other-downing beliefs*, which refer to negative global evaluations of others; and *awfulizing beliefs*, due to which daily-life adverse outcomes are seen as catastrophes. Some authors also recognized the role of rational beliefs of different kinds [29]. We limited our analysis on *unconditional self-acceptance* a concept that represents a rational view of the self.

We expect that teachers who adopt a custodial view on pupil control ideology will manifest a higher level of irrational beliefs, and in particular of other-related demandingness, often found among authoritarian teachers [34]. We also expect an inverse association between the teachers' preference for a custodial view and their level of unconditional self-acceptance, as the latter term is inversely correlated with irrational beliefs [35, 36].

**H2A.** *We expect that teachers with a custodial view on pupil control ideology will manifest a higher level of irrational beliefs.*

**H2B.** *We expect that teachers with a custodial view on pupil control ideology will manifest a higher level of other-directed demandingness as specific irrational beliefs.*

**H2C.** *We expect that teachers with a custodial view on pupil control ideology will manifest a lower level of unconditional self-acceptance.*

The third level of analysis of investigation linked the pupil control ideology to early maladaptive schemas from schema therapy [37]. The maladaptive schemas represent cognitive schemas, which ultimately synthesize our views of ourselves, others, and the world around us, and upon which beliefs and automatic thoughts are based. They are core beliefs in Beck's conceptualization [25]. Some of these maladaptive schemas are relevant for the custodial view on pupil control ideology. One such schema is *mistrust*, an expectation that someone will be mistreated by others, usually by intention. Its reverse state, trust, is a feature of a humanistic view. Another relevant schema is *unrelenting standards*, an expectation that one must meet unrealistic high standards. Such a perfectionistic perspective is more typical for a custodial view. *Entitlement* is another relevant schema because it refers to the expectation that the rule of reciprocity does not apply to himself/herself. Therefore, one can force his or her point of view and control the behavior of others, a feature that is typical for a custodial view. Last, but not least, *punitiveness* is another relevant schema referring to the expectation that people should be punished harshly for making mistakes, a perspective that departs from a humanistic view.

In line with the above expectations, the following hypotheses are derived.

**H3A.** *We expect that teachers with a custodial view on pupil control ideology will manifest a higher level of the mistrust schema.*

**H3B.** *We expect that teachers with a custodial view on pupil control ideology will manifest a higher level of the unrelenting standards schema.*

**H3C.** *We expect that teachers with a custodial view on pupil control ideology will manifest a higher level of the entitlement schema.*

H3D. *We expect that teachers with a custodial view on pupil control ideology will manifest a higher level of the punitiveness schema.*

Nonetheless, since a custodial view seems to be related to perfectionism tendencies, we complemented the investigation with the analysis of the relationship between the pupil control ideology and dimensions of perfectionism, as a multidimensional construct. We opted for an 8-dimension perfectionism structure proposed by Hill et al. [38], who integrates two popular models, the one proposed by Frost et al [39] and the one proposed by Hewitt & Flett [40]. Among the eight dimensions, we expected that a custodial view on pupil control ideology is more closely related to *Striving for excellence* (pursuing perfect results and high standards) and *High standards for others* (holding others to one's perfectionist ideals). Therefore, the last two research hypotheses were:

H4A. *We expect that teachers with a custodial view on pupil control ideology will score higher on Striving for Excellence dimension of perfectionism.*

H4B. *We expect that teachers with a custodial view on pupil control ideology will score higher on High Standards for Others dimension of perfectionism.*

## Method

### Participants and procedure

A convenience sample of 155 teachers (110 females, *M* age = 42.4, *SD* age = 9.3) from nineteen schools across three different Romanian counties was used in this study. Of this group, 77% were high school teachers, and 23% were middle school teachers. Teaching experience ranged from 1 to 40 years (*M* = 16.4, *SD* = 9.0). Participation was based on teachers' voluntary consent; no incentive was included. Each participant signed an Informed Consent Form before receiving the set of questionnaires. Data was collected, ensuring anonymity. The study was approved by the West University Human Research Ethics Committee.

### Instruments

The *Pupil Control Ideology Scale* (PCI) is a twenty-item measure on a five-point Likert scale (from strongly disagree to strongly agree) [8]. The higher the score, the more custodial the pupil control ideology is. The lower the scores, the more humanistic the pupil control ideology is. Sample items include "Pupils can be trusted to work together without supervision" (reversed) and "Pupils cannot perceive the difference between democracy and anarchy in the classroom." The PCI scale reliability in this study was.73.

*The Perfectionism Cognitions Inventory* [32] measures the frequency of perfectionistic automatic thoughts experienced in the last week. It consists of twenty-five items on a five-point Likert scale (from never to always). The higher the score, the higher the frequency of perfectionistic automatic thoughts experienced in the last week. Sample items include "I am too perfectionistic" and "I should never make the same mistake twice." The scale reliability in this study was.94.

*The Hostile Automatic Thoughts* (HAT) [33] measures the frequency of hostile automatic thoughts experienced in the last week that involves physical aggression, revenge, and derogation of others, using a five-point Likert scale (from never to always). We used a shorter version of this scale for the present study, after excluding the items expressing hostile thoughts that involve physical aggression (e.g., "I want to smack this person"). The higher the score, the higher the frequency of hostile thoughts experienced in the last week. Sample items include

"This person is a loser" for the derogation of other subscale and "This person needs to be taught a lesson" for the revenge subscale. The reliability coefficients in this study were.94 for the entire scale,.88 for the Derogation of others subscale, and.91 for the Revenge subscale.

*The Attitudes and Belief Scale (ABS 2)* is a 72-item self-report measure of rational and irrational beliefs, using a five-point Likert scale from strongly disagree to strongly agree [41]. We used a version that contains five irrational cognitive processes that could be summed up in a total score for irrationality. These processes are: demandingness (e.g., "I must do well at important things, and I will not accept it if I do not do well"), catastrophizing (e.g., "If loved ones or friends reject me, it is not only bad, but the worst possible thing that could happen to me"), frustration intolerance (e.g., "I can't stand being tense or nervous and I think tension is unbearable"), self-downing (e.g., "If I do not perform well at tasks that are very important to me, it is because I am a worthless bad person"), and other-downing (e.g., "If people treat me without respect, this indicates how bad they are in reality"). The higher the score, the higher the irrationality level. Whereas the scale also measures rational cognitive processes (e.g., preferences), we limited our focus on the irrational aspects. The reliability coefficients were.91 for the overall irrationality,.60 for demandingness,.59 for catastrophizing,.62 for frustration intolerance,.75 for self-downing, and.70 for other-downing.

*The Survey of Personal Beliefs (SPB)* [42] covers similar aspects as the ABS-2 measure. In this study, we used a 30-item brief version [43]. The SPB has the merit to disentangle the demandingness dimension into two dimensions: self-directed (e.g., "I definitely have to do a good job on all things I decide to do") and other-directed (e.g., "I believe that people definitely should not behave poorly in public"). The sub-scale reliability in this study was.76 for Self-Directed Should,.54 for Other-Directed Should,.72 for Awfulzing,.69 for Low Frustration Tolerance, and.67 for Self-Worth.

*The Unconditional Self-Acceptance (USAQ)* [44] is a 20-item measure of unconditional self-acceptance, using a seven-point Likert scale from almost always false to almost always true. Sample item includes "I feel worthwhile even if I am not successful in meeting certain goals that are important to me." The higher score indicates a higher level of unconditional self-acceptance. The scale reliability in this study was.72.

*The Young Schema Questionnaire*–the earlier version of Short Form 3 (YSQ-S3) [45] was used in this study. This version previously adapted in Romania [46] contains 114 items, covering 18 maladaptive schemas. The YSQ-S3 is a self-report measure using a six-point Likert scale, from "completely untrue of me" to "it describes me perfectly." The higher the score, the more present is that schema. The reliability for maladaptive schemas in this study ranged from 57 for Distrust / Abuse scale to.78 for Negativity / Pessimism scale.

*The Perfectionism Inventory Scale (PI)* [38] is a 59-item measure, using a five-point Likert scale from totally disagree to totally agree to assess a multidimensional form of perfectionism with eight dimensions. The sub-scales reliability in this study was.74 for Concern over Mistakes,.74 for High Standards for Others,.83 for Need for Approval,.58 for Organization,.76 for Planfulness,.89 for Perceived Parental Pressure,.62 for Rumination, and.61 for Striving for Excellence. Sample item includes "I get upset when other people do not maintain the same standards as I do" (high standards for others). The reliability coefficient for the overall level of perfectionism was.91.

## Data analysis strategy

Since the investigation of the association between the pupil control ideology and all measured dysfunctional beliefs will provide more than forty zero-order correlation coefficients (see Table 1), we employed some decisions to manage the risk for the family-wise error. First, we

**Table 1. Descriptive statistics and Pearson correlations between PCI and the other variables included in the study (N = 155).**

| Variables | M | SD | Potential Range | Actual Range | PCI r | PCI rp | Variables | M | SD | Potential Range | Actual Range | PCI r | PCI rp |
|---|---|---|---|---|---|---|---|---|---|---|---|---|---|
| **Automatic thoughts** | | | | | | | **Irrational beliefs** | | | | | | |
| Perfectionism | 38.3 | 21.7 | 0–100 | 2–93 | .20* | .22* | ABS Overall irrationality | 53.3 | 21.2 | 0–144 | 0–108 | .17* | .16* |
| Hostility | 31.1 | 9.9 | 19–95 | 19–76 | .19* | .18* | ABS Demandingness | 29.6 | 8.0 | 0–72 | 7–49 | .11 | .11 |
| Revenge | 12.6 | 5.1 | 9–45 | 9–43 | .05 | .02 | SPB Self-directed demands | 23.4 | 5.7 | 6–36 | 8–35 | .04 | .04 |
| Derogation of others | 18.5 | 6.0 | 10–50 | 10–40 | .28* | .27* | SPB Other-directed demands | 23.9 | 6.2 | 6–36 | 6–35 | .15 | .16* |
| **Early maladaptive schemas** | | | | | | | ABS Catastrophizing | 25.2 | 9.3 | 0–72 | 0–47 | .14 | .14 |
| Emotional deprivation | 9.0 | 4.3 | 5–30 | 5–23 | .15 | .10 | SPB Awfulizing | 20.4 | 5.8 | 6–36 | 6–33 | .01 | .03 |
| Abandonment / Instability | 8.2 | 3.5 | 5–30 | 5–19 | -.02 | -.05 | ABS Frustration intolerance | 26.8 | 10.9 | 0–72 | 2–48 | .09 | .12 |
| Mistrust / Abuse | 10.7 | 4.0 | 5–30 | 5–24 | .18* | .16* | SPB Frustration intolerance | 20.3 | 5.6 | 6–36 | 7–32 | .09 | .10 |
| Social isolation / Alienation | 8.2 | 3.4 | 5–30 | 5–22 | .03 | .00 | ABS Other-downing | 4.9 | 3.2 | 0–16 | 0–14 | .10 | .12 |
| Defectiveness / Shame | 6.8 | 3.1 | 5–30 | 5–22 | .05 | .01 | ABS Self-downing | 16.4 | 10.7 | 0–72 | 0–60 | .10 | .12 |
| Failure | 8.4 | 3.5 | 5–30 | 5–19 | .04 | .00 | SPB Conditional self-worth | 18.9 | 5.0 | 6–36 | 6–32 | .04 | .05 |
| Dependence / Incompetence | 8.8 | 3.3 | 5–30 | 5–20 | .07 | .02 | ABS Overall rationality | 44.8 | 23.3 | 0–144 | 0–122 | .03 | .05 |
| Vulnerability to harm/ illness | 7.3 | 3.4 | 5–30 | 5–25 | -.05 | -.10 | Unconditional self-acceptance | 89.6 | 10.9 | 20–140 | 68–121 | -.20* | -.23* |
| Enmeshment/undeveloped self | 9.6 | 3.5 | 5–30 | 5–19 | .11 | .09 | **Perfectionism dimensions** | | | | | | |
| Subjugation | 9.5 | 4.6 | 5–30 | 5–23 | .00 | -.03 | Overall perfectionism | 24.1 | 4.4 | 8.0–40.0 | 14.4–37.0 | .21* | .22* |
| Self-sacrifice | 18.9 | 6.2 | 5–30 | 5–30 | .05 | .08 | Concern over mistakes | 2.4 | 0.7 | 1.0–5.0 | 1.0–4.8 | .16* | .17* |
| Emotional inhibition | 12.8 | 4.3 | 5–30 | 5–25 | .16* | .18* | High standards for other | 2.8 | 0.7 | 1.0–5.0 | 1.3–4.9 | .18* | .18* |
| Unrelenting standards | 16.3 | 4.1 | 5–30 | 5–24 | .23* | .23* | Need for approval | 2.5 | 0.8 | 1.0–5.0 | 1.0–4.9 | .12 | .12 |
| Entitlement / Grandiosity | 14.1 | 4.7 | 5–30 | 5–25 | .17* | .21* | Organization | 4.0 | 0.9 | 1.0–5.0 | 1.5–5.0 | .08 | .11 |
| Insufficient self-control | 11.9 | 3.7 | 5–30 | 5–23 | .20* | .22* | Parental pressure | 2.5 | 0.9 | 1.0–5.0 | 1.0–5.0 | .11 | .13 |
| Approval seeking | 33.1 | 9.2 | 14–84 | 14–58 | .19* | .16* | Planfulness | 3.9 | 0.6 | 1.0–5.0 | 1.9–5.0 | .14 | .14 |
| Negativity / Pessimism | 21.8 | 7.7 | 11–66 | 11–44 | .13 | .13 | Rumination | 2.7 | 1.0 | 1.0–5.0 | 1.0–4.9 | .14 | .14 |
| Punitiveness | 36.5 | 12.0 | 14–84 | 14–83 | .12 | .14 | Striving for excellence | 3.3 | 1.0 | 1.0–5.0 | 1.5–4.7 | .17* | .19* |

* for $p$s = .05 or lower (two-tailed).

Reference values: $|r| \geq .16$, $p < .05$ and $|r| \geq .21$, $p < .01$, two-tailed tests; PCI $r$–zero-order correlation between the pupil control ideology (custodial view for high scores) and the other variables; PCI $rp$–partial correlation (controlling for gender, years of teaching, school level, and residential area); descriptive statistics values for the PCI variable are $M = 65.1$ $SD = 8.6$, with a theoretical range of scores between 20 and 100.

treated all our unidirectional hypotheses as primary outcomes on which our focus was laid on given their theoretical rationale. The remaining associations represented post hoc analyses and were treated as secondary findings resulted from a bidirectional exploratory approach. We were also more conservative for these exploratory analyses by adjusting the p-value for a significant result from .05 to .01.

Another post hoc data analysis strategy was to complement the zero-order correlation coefficients with partial correlation coefficients when controlling for four demographic variables: teacher's gender, years of experience in teaching, the school level (lower secondary vs. upper secondary schools), and the school area (urban vs. rural). This strategy is indicative of the stability of the results, with a particular focus on those referring to hypotheses-testing.

## Results

We first examined the correlations that addressed the first set of hypotheses (1A and 1B) that investigated the relationship between teachers' pupil control ideology and specific automatic thoughts. We found empirical support for both hypotheses. Teachers with a custodial view on pupil control ideology have perfectionistic thoughts more frequently—$r$ (153) = .20, $p$ = .006, one-tailed test. They also have hostile thoughts more frequently—$r$ (153) = .19, $p$ = .008,

one-tailed test. Post-hoc analyzes showed that teachers with a custodial view are more likely to experience derogation of others thoughts. Likewise, both results remain statistically significant when controlling for gender, years of teaching, school level, and school area (see Table 1 for details).

Next, we examined the correlations that addressed the second set of hypotheses (2A, 2B, and 2C). We found empirical support for all three hypotheses. Teachers with a custodial view on pupil control ideology endorsed more irrational beliefs—$r$ (153) = .17, $p$ = .019, one-tailed test, and more other-directed demandingness beliefs—$r$ (153) = .15, $p$ = .029, one-tailed test. They were less likely to manifest unconditional self-acceptance—$r$ (153) = -.19, $p$ = .008, one-tailed test. Post-hoc analyses revealed no other significant connection between various types of irrational beliefs and the pupil control ideology ($p$ >.05, two-tailed test). Likewise, all three hypothesizes remain statistically significant when controlling for gender, years of teaching, school level, and school area (see Table 1 for details).

Next, we examined the correlations that addressed the third set of hypotheses (3A, 3B, 3C, and 3D) that investigated the relationship between pupil control ideology and specific early maladaptive schemas. We found empirical support for three out of four hypotheses. Teachers with a custodial view were more likely to show the mistrust schema—$r$ (153) = .18, $p$ = .012, one-tailed test, were more likely to show the unrelenting standards / hypercriticism schema—$r$ (153) = .23, $p$ = .002, one-tailed test, and were more likely to show the entitlement schema—$r$ (153) = .17, $p$ = .017, one-tailed test. Data did not support one hypothesis—$r$ (153) = .12, $p$ = .071, one-tailed test, suggesting no relationship between the pupil control ideology and the punitiveness schema. Post-hoc analyses revealed additional statistically significant associations between the pupil control ideology and the following maladaptive schemas: insufficient self-control, approval-seeking, and emotional inhibitions. A similar pattern of statistically significant results was obtained when controlling for the four demographic variables (see Table 1).

Finally, we inspected the correlations that addressed the last set of hypotheses (4A and 4B) that examined the relationship between pupil control ideology and specific dimensions of perfectionism. We found empirical support for both hypotheses. Teachers with a custodial view on pupil control ideology scored higher on Striving for Excellence dimension of perfectionism—$r$ (153) = .17, $p$ = .016, one-tailed test, and scored higher on High Standards for Others dimension of perfectionism—$r$ (153) = .18, $p$ = .014, one-tailed test. Post-hoc analyses found that a custodial view on pupil control ideology also correlates with the global level of perfectionism, and with the concern over mistakes dimension of perfectionism. The same pattern of statistically significant results was obtained when controlling for the four demographic variables (see Table 1).

## Discussion

We investigated whether the teacher's pupil control ideology is associated with various dysfunctional beliefs. Our findings suggest that teachers who hold a custodial view on the pupil control ideology endorse more dysfunctional beliefs than teachers who hold a humanistic view on the pupil control ideology.

Teachers who adopt a custodial view tend to be more psychologically inflexible as they show a higher level of perfectionism. They apply high demands on themselves by striving for excellence and having high self-internalized expectations—unrelenting standards. They also apply high demands on others through high standards for others and a higher level of other-directed demandingness. These teachers also possess relevant dysfunctional beliefs for their interaction with others. They have endorsed more often the schema of mistrust, suggesting they expect pupils to take advantage and neglect their academic role in a loose classroom

disciplinary policy. However, such a view does not generalize on all pupils. Instead, teachers with a custodial view on pupil control ideology are more likely to judge and react harshly on pupils who misbehave. In such circumstances, they are more prone to use derogation of others thoughts and are less able to distinguish the evaluation of the pupils from the evaluation of their behavior. They also tend to impose their views on others and to focus less on others' needs as they manifest entitlement schema.

To the best of our knowledge, no research has explicitly focused on the relationship between the teacher's pupil control ideology and the types of dysfunctional beliefs they endorse. While previous research highlighted the detrimental role of a custodial view on pupil control ideology for the quality of the teacher-student relationship, the present findings revealed some cognitive vulnerabilities that are associated with such a custodial view. Such findings contribute to the existing literature by providing reasons for why custodial teachers encounter a high prevalence of burnout and distress. Our results are consistent with previous findings [47, 48]. The former authors showed that the unrelenting standards schema is a predictor of burnout, whereas the latter authors argued that professionals who experience difficulties in setting boundaries with their clients experience a higher level of burnout. Custodial teachers, who manifest both high-expectations (unrelenting standards schema) and difficulties in setting boundaries with their pupils (mistrust, entitlement, and insufficient self-control schemas), are even more likely to experience burnout.

More importantly, the current research provides some clues on how school counselors and other professionals might intervene to improve the teacher-student interaction and to reduce the level of stress among teachers. Previous research was limited to addressing educational roots for the pupil control ideology. Teacher's approach to education (e.g., traditionalism, progressivism) was the most relevant predictor for the teacher's pupil control ideology [49]. The present research provides a cognitive-behavioral perspective on possible psychological antecedents for the pupil control ideology.

The empirical results from this study should be considered in light of some limitations. First, the cross-sectional design for the present research prevents us from empirically establishing whether such cognitive correlates are indeed antecedents for the pupil control ideology teachers adopt. Likewise, in the absence of a longitudinal design, we cannot analyze these relationships in their dynamics. Some dysfunctional beliefs (e.g., derogation of others) could result from the emotional distress lived by teachers, caused by their perfectionistic and rigid demands. Second, some of the scales used to measure dysfunctional beliefs had low reliability, with Cronbach's alpha lower than .70. These values signal a significant level of measurement error in data, leading to a decrease in the statistical power to address some of present study hypotheses. However, despite the attenuation of the effect size, most primary research hypothesizes were supported by data.

The relevance of several dysfunctional beliefs for the teacher's pupil control ideology suggests that future research could test the effectiveness of cognitive-behavioral interventions to orient teacher's pupil control ideology in a more adaptive way.

To sum up, this study has increased our knowledge regarding how several dysfunctional beliefs are correlated with teachers' view on exerting control over pupils in the classroom. The overall picture is that teachers who adopt a custodial view on the pupil control ideology are also more likely to manifest a dysfunctional view on pupils who misbehave, perceiving them as adversaries who threaten their rigid and/or perfectionistic expectations. Future studies, and more specifically, longitudinal studies, are needed to investigate this topic because of multiple benefits. The expected gains are not limited to the increase of the well-being of teachers, but also to the increase in the quality of the teacher-student relationship, from which both teachers and students might benefit.

## Author Contributions

**Conceptualization:** Elena Mirela Samfira, Florin Alin Sava.

**Data curation:** Elena Mirela Samfira.

**Investigation:** Elena Mirela Samfira.

**Methodology:** Elena Mirela Samfira.

**Resources:** Elena Mirela Samfira.

**Writing – original draft:** Florin Alin Sava.

**Writing – review & editing:** Elena Mirela Samfira.

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
