## [Decision Letter · Decision Letter 0]

27 Jul 2020

PONE-D-19-32116

Cognitive-behavioural correlates of pupil control ideology

PLOS ONE

Dear Dr. Florin,

Thank you for submitting your manuscript to PLOS ONE. After careful consideration, we feel that it has merit but does not fully meet PLOS ONE’s publication criteria as it currently stands. Therefore, we invite you to submit a revised version of the manuscript that addresses the points raised during the review process.

The reviewers have raised concerns about the methodology and design of the study, including issues with sample size. The reviewers' comments can be viewed in full below.

We look forward to receiving your revised manuscript.

Kind regards,

Natasha McDonald, PhD

Associate Editor

PLOS ONE

Journal Requirements:

Additional Editor Comments (if provided):

Reviewers' comments:

Reviewer's Responses to Questions

**Comments to the Author**

1. Is the manuscript technically sound, and do the data support the conclusions?

Reviewer #1: Partly

Reviewer #2: Yes

Reviewer #3: Partly

2. Has the statistical analysis been performed appropriately and rigorously? 

Reviewer #1: No

Reviewer #2: Yes

Reviewer #3: No

3. Have the authors made all data underlying the findings in their manuscript fully available?

Reviewer #1: Yes

Reviewer #2: Yes

Reviewer #3: No

4. Is the manuscript presented in an intelligible fashion and written in standard English?

Reviewer #1: Yes

Reviewer #2: Yes

Reviewer #3: Yes

5. Review Comments to the Author

Reviewer #1: Based on a sample with N = 155 teachers, this study investigated correlations between teachers’ pupil-control ideology and different types of beliefs that are suggested to be relevant in cognitive-behavioral therapy. Results of correlation analyses revealed that teachers with control ideology had more dysfunctional beliefs than the ones with a humanistic ideology.

I appreciate the general idea guiding this paper and find merit in investigating how teachers’ beliefs shape their relationships with students. However, I was missing a compelling theoretical rationale and empirical evidence supporting your idea to focus on pupil-control ideology. Because no data on how teacher beliefs play out in the classroom (or perhaps for teacher well-being) is presented, I am concerned that this contribution does little to answer the question of how we can help teachers improve the quality of their relationships with students. In addition, this cross-sectional study only relies on teacher self-reports of various types of profession-specific (i.e., pupil-control ideology) and general beliefs (i.e., dysfunctional beliefs derived from cognitive-behavioral therapy) and merely investigates correlations. The pending question here is: Why does it matter? Why do we need to know how pupil-control ideology correlates with the many beliefs that seem to measure conceptually overlapping constructs? In addition to these general concerns, I will provide some more detailed suggestions below.

Abstract

I am not sure that pupil control ideology is a well-known term so you may have to clarify its meaning in the abstract already. Likewise, I imagine it will be difficult to know for readers with a background in education what “personal beliefs originated from cognitive-behavioral therapy” are.

Introduction

In general, I think the manuscript would profit from more precise definitions of key terms.

You claim that teachers’ pupil control ideology is a “core construct for the quality of teacher-pupil relationship”, but where is the evidence supporting this strong claim? In my view, you need to be much more specific in terms of the (empirical) relevance of pupil control ideology for teacher-student relationships, teacher well-being or other key outcomes in the teaching profession. If you cannot support the relevance of pupil control ideology, I am quite skeptical that educational research and practice could profit from the insights your study provides. You also may want to provide more detail in this regard.

Given the central role of teacher-student relationships in your manuscript, I would like to see a clear definition. On p. 4 you cite a few studies which are supposed to underpin the relevance of pupil control ideology for the teacher-student relationship, but I am not convinced that things like corporal punishment, teaching efficacy, and academic optimism are the same as a high-quality teacher-student relationship.

A more detailed definition of automatic thoughts, schemas, and irrational beliefs would be helpful. Moreover, I was wondering how pupil control ideology and the beliefs from cognitive-behavioral therapy are actually distinct from one another. That is, could you also view pupil control ideology as a profession-specific type of these beliefs? Due to the strong theoretical overlap between all of the constructs you are measuring, I imagine that factor analyses are necessary to get a clearer impression of how the things you measure are distinct from one another or rather very similar.

Overall, I found the presentation of hypotheses not easy to follow. For instance, I did not understand whether “unconditional self-acceptance” is a superordinate concept for demandingness beliefs, frustration tolerance beliefs, etc. or whether these terms are independent of each other. Furthermore, I found it confusing that perfectionisms was mentioned in the first few hypotheses and then again at the very end of the introduction.

Method and Results

Because you analyze the multiple dimensions of the perfectionism scale separately, you should mention each subscale in the description of the instruments and report Cronbach’s alpha for each of them.

Why were different scales used for measuring the same construct (e.g., ABS, SPB, and YSQ, two perfectionism scales)?

As mentioned earlier, I think factor analyses would be necessary to investigate whether the various different scales you included are empirically distinct. A correlation table for all included measures would add to this.

There is a large number of statistical tests. Thus, the authors may want to consider adjusting alpha to avoid false positive findings.

In Table 1 it is difficult to evaluate whether the mean values are in a high or low range. Moreover, it is difficult to see at first glance which of the correlations are statistically significant and which are not.

Minor points

A reference is missing in line 83/84

All statistical symbols should be printed in italics

The reference list should be checked for consistent spelling (e.g., capitalization of first letters)

Reviewer #2: The topic selection of this paper has certain practical significance. The paper provides some innovative results. The literature review is sufficient. The method is reasonable. The discussion was in-depth.

Reviewer #3: Thank you for the opportunity to review, “Cognitive Behavioral Correlates of Pupil Control Ideology” for PLOS ONE. This study examined the relations between pupil control, defined on a spectrum from humanistic to custodial, and teachers’ perceptions of perfectionism and hostile automatic thoughts. The authors found that teachers with more custodial perceptions of control had higher ratings of perfectionism and hostile thoughts, but no relation was found for punitiveness schema. Overall, the manuscript is generally clear and the study provides some interesting insights into the relations of the constructs. That being said, there are few concerns. First, it is unclear if the hostile thoughts measure was adapted to refer to hostile thoughts towards students. Although general thoughts of hostility towards others are potentially correlated with thoughts about students, there is a difference and that should be made explicit.

Second, a brief data analysis section in the method section should be added. In particular, the authors should discuss using one-tailed tests and, per Table 1, two-tailed test and justify each. Relatedly, the author should consider the likelihood of family-wise error rates given the series of correlations of related constructs. The author also needs to provide information about how they scaled the measure scores and the min/max values. Lastly, the reliance on correlations does not necessarily address the researchers’ primary interests and do not control for potentially relevant covariates, including years teaching. The data provide opportunities for some mediation analyses (e.g., do irrational beliefs mediate the relation between PCI and Hostility) and, more importantly, comparison of groups. The authors could create groups of humanistic and custodial teachers and then use t-tests, or ideally, a regression or ancova model controlling for teacher characteristics.

Page 6- line 101- there should be “ before My Work

Page 8- name the author for reference 28 as the sentence says, “proposed by…”

Page 9, line 175- consider “incentive” instead of “reward”

Page 10- reliability coefficients should be reported for the Survey of Personal Beliefs.

Page 11- reliability coefficients should be reported for the Youth Schema Questionnaire.

6. PLOS authors have the option to publish the peer review history of their article (what does this mean?). If published, this will include your full peer review and any attached files.

Reviewer #1: No

Reviewer #2: **Yes: **Hai Zhang

Reviewer #3: No

---

## [Author Response · Author response to Decision Letter 0]

21 Sep 2020

We attached a file entitled response to reviewers that respond to specific comments made by reviewers and editor.

---

## [Decision Letter · Decision Letter 1]

12 Nov 2020

PONE-D-19-32116R1

Cognitive-behavioural correlates of pupil control ideology

PLOS ONE

Dear Dr. Florin,

Thank you for submitting your manuscript to PLOS ONE. After careful consideration, we feel that it has merit but does not fully meet PLOS ONE’s publication criteria as it currently stands. Therefore, we invite you to submit a revised version of the manuscript that addresses the points raised during the review process.

We look forward to receiving your revised manuscript.

Kind regards,

Nicholas Gage

Academic Editor

PLOS ONE

Additional Editor Comments (if provided):

Thanks for the opportunity to serve as guest editor for this manuscript. Overall, the authors have done an excellent job revising their manuscript. That being said, I would like the authors to address the first two concerns noted by Reviewer #1, namely a little more justification for the research question and, importantly, noting the reliability concerns in the limitations section.

Reviewers' comments:

Reviewer's Responses to Questions

**Comments to the Author**

1. If the authors have adequately addressed your comments raised in a previous round of review and you feel that this manuscript is now acceptable for publication, you may indicate that here to bypass the “Comments to the Author” section, enter your conflict of interest statement in the “Confidential to Editor” section, and submit your "Accept" recommendation.

Reviewer #1: (No Response)

Reviewer #2: All comments have been addressed

2. Is the manuscript technically sound, and do the data support the conclusions?

Reviewer #1: Yes

Reviewer #2: Yes

3. Has the statistical analysis been performed appropriately and rigorously? 

Reviewer #1: Yes

Reviewer #2: Yes

4. Have the authors made all data underlying the findings in their manuscript fully available?

Reviewer #1: Yes

Reviewer #2: Yes

5. Is the manuscript presented in an intelligible fashion and written in standard English?

Reviewer #1: Yes

Reviewer #2: Yes

6. Review Comments to the Author

Reviewer #1: First, I would like to thank the authors for the thorough revision of their manuscript. In particular, I value the added detail on dysfunctional beliefs and inclusion of clearer definitions. Nonetheless, I still find myself unconvinced of the theoretical and practical relevance of the research questions. To make the paper more convincing for the readers, I suggest you explain why these insights are of interest. Simply saying that the correlation between custodial views and irrational beliefs has not been investigated is not really an argument from my perspective.

Furthermore, I noticed that there were some scales with reliabilities with α < .70 (i.e., some subscales of ABS 2, SPB, Young Schema Questionnaire, and Perfectionism Inventory). This is an issue you may want to address in the discussion.

Finally, I still think that factor analyses are necessary and the authors’ argument that an overlap between different dysfunctional cognitions layers is expected underpins this thought rather than dispelling it. Imagine that, for example, ten of the dysfunctional beliefs are correlated by r = .70 and load on the same factor. Then, one would argue that they measure the same underlying construct. Hence, finding ten statistically significant correlations between the ten dysfunctional beliefs and custodial orientations would seem impressive, when in fact you simply investigated ten “variations” of one and the same construct making it little surprising that all of them are associated with custodial orientations in similar ways.

Reviewer #2: (No Response)

7. PLOS authors have the option to publish the peer review history of their article (what does this mean?). If published, this will include your full peer review and any attached files.

Reviewer #1: No

Reviewer #2: No

---

## [Author Response · Author response to Decision Letter 1]

10 Jan 2021

Thank you for the opportunity to revise the manuscript entitled “Cognitive-behavioral correlates of pupil control ideology” (PONE-D-19-32116R1).

In the following lines, you can find how we addressed each of these three points that remained after the first revision. 

***

Editor of this manuscript: The authors have done an excellent job revising their manuscript. That being said, I would like the authors to address the first two concerns noted by Reviewer #1, namely a little more justification for the research question and, importantly, noting the reliability concerns in the limitations section.

Reviewer #1: First, I would like to thank the authors for the thorough revision of their manuscript. In particular, I value the added detail on dysfunctional beliefs and inclusion of clearer definitions. Nonetheless, I still find myself unconvinced of the theoretical and practical relevance of the research questions. To make the paper more convincing for the readers, I suggest you explain why these insights are of interest. Simply saying that the correlation between custodial views and irrational beliefs has not been investigated is not really an argument from my perspective.

#1 In the revised version, we extended the study purpose by including one additional page. We explained: (a) why the pupil control ideology is worth being studied from a psychological perspective, and not only from an educational sciences perspective; (b) using the cognitive-behavioral framework adds practical value in addressing this topic. I inserted below the text that was introduced to address the concern of Reviewer 1.

Addressing this research question is relevant for both theoretical and practical reasons. From a theoretical perspective, the study contributes to enriching the literature on pupil control ideology underpinnings. As already underlined, most previous studies have focused on antecedents and consequences of pupil control ideology from an educational science perspective. There were significantly fewer studies that aimed at addressing this topic from a psychological perspective. Teachers strive to achieve a sense of competence and success. However, their efforts are often undermined by the lack of personal control and unpredictability in a classroom context, leading to stress or burnout. Most psychological studies have focused on depicting the custodial teacher personality profile (e.g., more authoritarian, less open, etc.). Such findings could help determine who the teachers at-risk for manifesting a problematic teacher-student interaction are. However, they are less useful for an intervention purpose because personality traits are stable and difficult to change. Such findings are also less relevant to the aim of decreasing the level of stress among teachers. 

That is why examining the link between dysfunctional beliefs and the pupil control ideology is of more practical value. The core of any cognitive-behavioral intervention consists of replacing dysfunctional beliefs with more functional ones. Such interventions could provide a new avenue to alter pupil control ideology indirectly by changing the correlated dysfunctional beliefs. Changing teachers' cognitions on aspects such as perfectionistic cognitions or their demandingness beliefs will most likely influence their interaction with students by altering the amount of control they exert in the classroom context. Likewise, addressing teachers' cognitions that could impair interpersonal interaction, such as hostility or entitlement, will change teachers' beliefs about pupils with whom they interact daily, towards perceiving a less threatening classroom environment. Thus, our study opens an avenue towards potential cognitive-behavioral interventions in the school setting by first gathering evidence on the association between the pupil control ideology and specific dysfunctional beliefs. Such interventions have the potential of leading to multiple direct and indirect benefits regarding the classroom climate, classroom management, and teachers and pupils' well-being.

***

Reviewer #1: Furthermore, I noticed that there were some scales with reliabilities with α < .70 (i.e., some subscales of ABS 2, SPB, Young Schema Questionnaire, and Perfectionism Inventory). This is an issue you may want to address in the discussion.

Thank you for bringing up this issue. We delineated two main limitations of the current study, and we changed the text accordingly in the discussion section. I inserted the relevant paragraph below.

The empirical results from this study should be considered in light of some limitations. First, the cross-sectional design for the present research prevents us from empirically establishing whether such cognitive correlates are indeed antecedents for the pupil control ideology teachers adopt. Likewise, in the absence of a longitudinal design, we cannot analyze these relationships in their dynamics. Some dysfunctional beliefs (e.g., derogation of others) could result from teachers' emotional distress, caused by their perfectionistic and rigid demands. Second, some of the scales used to measure dysfunctional beliefs had low reliability, with Cronbach's alpha lower than .70. These values signal a significant level of measurement error in data, leading to a decrease in the statistical power to address some of the present study hypotheses. However, despite the attenuation of the effect size, most primary research hypothesizes were supported by data.

***

Reviewer #1: Finally, I still think that factor analyses are necessary and the authors’ argument that an overlap between different dysfunctional cognitions layers is expected underpins this thought rather than dispelling it. Imagine that, for example, ten of the dysfunctional beliefs are correlated by r = .70 and load on the same factor. Then, one would argue that they measure the same underlying construct. Hence, finding ten statistically significant correlations between the ten dysfunctional beliefs and custodial orientations would seem impressive, when in fact you simply investigated ten “variations” of one and the same construct making it little surprising that all of them are associated with custodial orientations in similar ways.

Conducting such an analysis in our study is not recommended, either theoretically or empirically. 

From a cognitive-behavioral perspective, there is no rationale to put in the same basket core beliefs, intermediate dysfunctional beliefs, and automatic thoughts as they differ in their stability, situational consistency, level of abstraction, and utility. That is why they are organized hierarchically in different layers of cognition. In the same way, one will not mix items with facets scores and domains scores when conducting factorial analyses in the personality field, for example. 

The very high correlation between any two constructs is not sufficient to disregard one of the constructs. For instance, depression and anxiety correlate highly (around .70), and this did not lead to the suggestion of dropping out one of these two constructs. Nor in the case of low self-esteem, despite its high correlation with depression. Therefore, two strongly correlated constructs could still keep their relevance, based on their discriminative validity and theoretical relevance, despite their high correlation. Take for example, the case of the ABS-2 instrument. All scales included in the instrument are highly correlated (in the range of .50 - .70). Despite the high correlation among each other, each scale is good predictor for a specific negative emotion (e.g. the low frustration scale for anger, the self-downing scale for sadness, the awfulizing scale for anxiety). Therefore, from a measurement perspective, all scales used in the current study are taken from the existing literature, they are not new scales. Interested readers could refer to their original validation studies and subsequent methodological studies. 

From a data sample perspective. Our sample is based on data gathered from approximately 150 participants. It is improbable to find a stable factorial solution in an analysis that involves about 40 variables (scales). Likewise, very high correlations were extremely uncommon in our sample. Out of 946 pairs of correlations at the scales level, only 7 (<1%) were within the range of .70 - .75, and only 36 (3,8%) were in the range of .60 to .69.

From a factorial analysis perspective, we know that a factorial solution is strongly dependent on the input variables. For instance, addressing the factorial solution of the perfectionism scale at the item-level reveals perfectionism's multidimensional nature. Dropping the item-level and conducting a factorial analysis using only the scale scores as input leads to the extraction of a single factor, the overall perfectionism level. Therefore, the factorial solution is highly dependable on input (variables to be analyzed). 

In the absence of a theory-driven or data-driven rationale to conduct such a factorial analysis, such an endeavor is likely to remain just a technical exercise. Therefore, we did not include such an analysis in the manuscript. First of all, because it would address a different purpose than the aim of the current manuscript. Second, because it is not theoretically and methodologically supported to mix automatic thoughts, intermediate beliefs, and core beliefs.

***

To conclude, we would like to thank both Reviewer 1 and Reviewer 2 for their appreciation and suggestions! We hope that the research question is now better justified in terms of its relevance and that the study's limitations are better addressed. Therefore, we hope that the manuscript's second revision has led to an improved version that the previous version of the manuscript.

---

## [Editor Report · Decision Letter 2]

27 Jan 2021

Cognitive-behavioural correlates of pupil control ideology

PONE-D-19-32116R2

Dear Dr. Florin,

We’re pleased to inform you that your manuscript has been judged scientifically suitable for publication and will be formally accepted for publication once it meets all outstanding technical requirements.

Kind regards,

Nicholas Gage

Guest Editor

PLOS ONE

Additional Editor Comments (optional):

The authors have addressed all of the reviewers concerns and I am recommending publication at this time. Well done.
---

## [Editor Report · Acceptance letter]

1 Feb 2021

PONE-D-19-32116R2 

Cognitive-behavioral correlates of pupil control ideology 

Dear Dr. Sava:

I'm pleased to inform you that your manuscript has been deemed suitable for publication in PLOS ONE. Congratulations! Your manuscript is now with our production department. 

Kind regards, 

on behalf of

Dr. Nicholas Gage 

Guest Editor

PLOS ONE